# Ion mobility collision cross-section atlas for known and unknown metabolite annotation in untargeted metabolomics

Zhiwei Zhou [1,2], Mingdu Luo[1,2], Xi Chen[1,2], Yandong Yin[1], Xin Xiong[1], Ruohong Wang[1,2] & Zheng-Jiang Zhu [1✉]

The metabolome includes not just known but also unknown metabolites; however, metabolite annotation remains the bottleneck in untargeted metabolomics. Ion mobility – mass spectrometry (IM-MS) has emerged as a promising technology by providing multi-dimensional characterizations of metabolites. Here, we curate an ion mobility CCS atlas, namely AllCCS, and develop an integrated strategy for metabolite annotation using known or unknown chemical structures. The AllCCS atlas covers vast chemical structures with >5000 experimental CCS records and ~12 million calculated CCS values for >1.6 million small molecules. We demonstrate the high accuracy and wide applicability of AllCCS with medium relative errors of 0.5–2% for a broad spectrum of small molecules. AllCCS combined with in silico MS/MS spectra facilitates multi-dimensional match and substantially improves the accuracy and coverage of both known and unknown metabolite annotation from biological samples. Together, AllCCS is a versatile resource that enables confident metabolite annotation, revealing comprehensive chemical and metabolic insights towards biological processes.

[1] Interdisciplinary Research Center on Biology and Chemistry, Shanghai Institute of Organic Chemistry, Chinese Academy of Sciences, 200032 Shanghai, People's Republic of China. [2] University of Chinese Academy of Sciences, 100049 Beijing, People's Republic of China. ✉email: jiangzhu@sioc.ac.cn

Untargeted metabolomics enables comprehensive measurements of a significant number of metabolites in complex systems, and identifies the accrued metabolic changes with physiological and pathological status, such as diseases[1,2]. Metabolites in the metabolome include knowns and unknowns generated from biotransformation of endogenous and exogenous compounds, and have a vast diversity of chemical structures[3]. Metabolite identification remains the central bottleneck in liquid chromatography—mass spectrometry (LC–MS)-based untargeted metabolomics[3–5]. The standard strategy for metabolite identification is to match accurate mass and tandem mass spectra (MS/MS or MS2) with standard spectral libraries (e.g., METLIN[6], MASSBANK[7], and NIST) and/or in-silico predicted MS/MS spectra[8]. However, standard spectral libraries suffer from the limited coverage, while the in-silico prediction lacks high accuracy[4]. Other bioinformatic approaches (e.g., GNPS[9], MetDNA[10]) also use MS2 spectra and molecular networking algorithms for metabolite annotations. All of these strategies require unique and high quality of experimental MS2 spectra. However, low molecular-weight metabolites usually have very sparse MS2 spectra and lack characteristic product ions for structural elucidation[4]. Some metabolite isomers share highly similar MS2 spectra. Many experimental factors, such as high sample complexity, low concentration and co-elution of isobaric and isomeric metabolites, present challenges to acquire high quality of MS2 spectra[4]. In addition, annotation of unknown metabolites with new chemical structures is still a challenge in untargeted metabolomics[3,11]. These issues cause low coverage and high false-positive rate of metabolite annotation, suggesting that other physiochemical properties should be developed for metabolite annotation.

Recently, ion mobility–mass spectrometry (IM–MS) has emerged as a promising technique for untargeted metabolomics by providing multi-dimensional separation and high selectivity[12–15]. Importantly, ion mobility can rapidly separate metabolite ions based on their differences in rotationally averaged surface area or collision cross-section (CCS)[16,17]. It enables to distinguish the isomeric metabolites that commonly exist in biological samples[18–21]. Unlike retention time (RT) and MS/MS spectra that are prone to be affected by many experimental factors, CCS is highly reproducible across instruments and labs, and it is much more feasible to be standardized[16,22]. The IM-derived CCS value is a unique physiochemical property to improve the accuracy of metabolite annotation. Significant efforts have been made to curate large-scale experimental and calculated CCS databases[23]. For example, Baker group[24] and McLean group[25,26] measured chemical standards to construct experimental CCS databases with >1000 CCS values. Nevertheless, these CCS resources are reported in quite different formats, and lack appropriate procedures and tools for data collection, collation, standardization, and sharing. Our group and others developed machine-learning-based prediction (e.g., MetCCS[27,28], LipidCCS[29], DeepCCS[30]) and quantum chemistry-based theoretical calculation (e.g., ISiCLE[31]) approaches to generate large-scale CCS values for metabolites, lipids and other compounds. Coupling IM–MS with LC separation and data-independent or data-dependent MS/MS techniques (e.g., MS$^E$, AIF, and PASEF) enables simultaneous acquisition of four-dimensional metabolomics data within one analysis, including MS1, RT, CCS, and MS/MS[32,33]. However, limited studies have integrated multi-dimensional properties in IM–MS towards the large-scale annotation of both known and unknown metabolite in untargeted metabolomics[11].

Here, we curate an ion mobility CCS atlas, namely, AllCCS, to embrace both experimental and predicted CCS values, and develop an integrated multi-dimensional match strategy to enable annotation of both known and unknown metabolites in IM–MS-based untargeted metabolomics (Fig. 1). The AllCCS atlas includes >5000 experimental CCS records and ~12 million predicted CCS values for >1.6 million compounds. The newly optimized machine-learning-based prediction utilizes a large training dataset with high diversity of chemical structures and a representative structure similarity (RSS) score to evaluate the accuracy

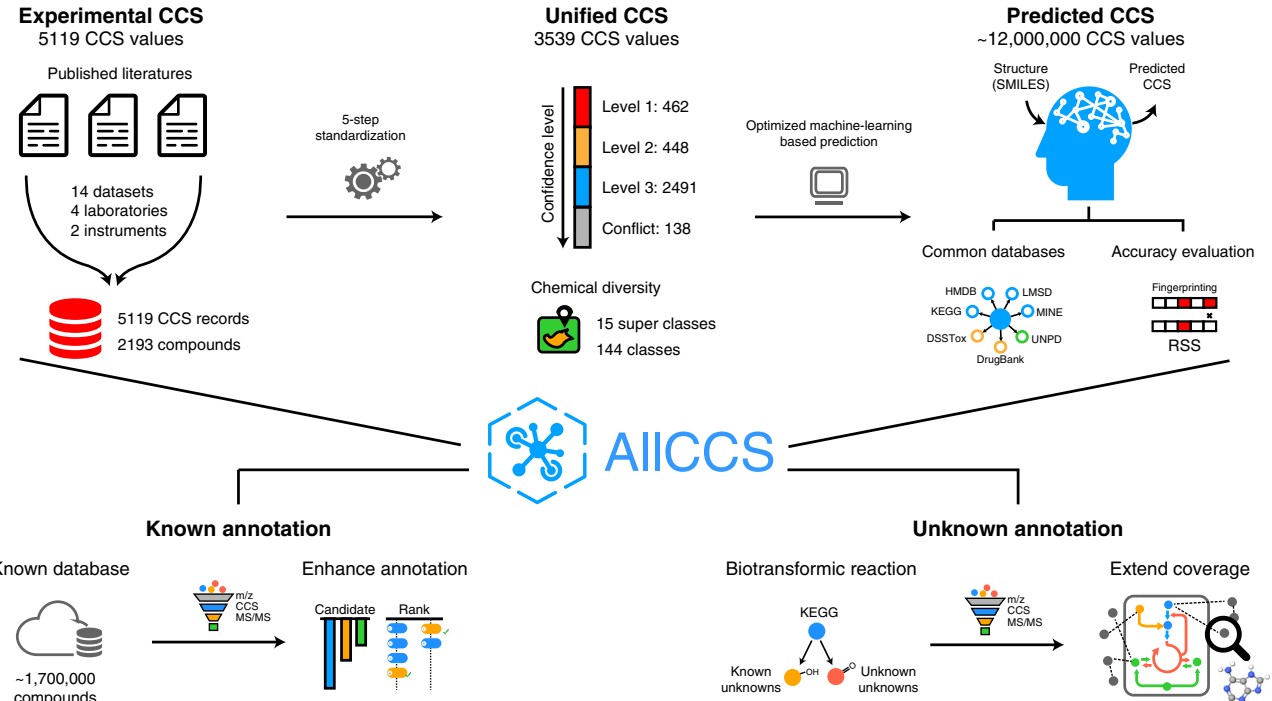

**Fig. 1 Overview of AllCCS atlas and annotation of known and unknown metabolites.** The AllCCS atlas hosts 5119 experimental CCS records, 3539 unified CCS values, and ~12 million predicted CCS values for ~1.7 million compounds. AllCCS can be integrated with in-silico MS/MS spectra to enable the multi-dimensional annotation for known and unknown metabolites in untargeted metabolomics.

of predicted CCS values. Our data shows that AllCCS outperforms other CCS calculation tools in terms of both coverage and accuracy. We further demonstrate that the use of AllCCS atlas and/or in-silico MS/MS spectra improves the annotation performances for both known and unknown metabolites in untargeted metabolomics. Taken together, AllCCS atlas is a valuable and unique resource to support IM–MS-based multi-dimensional metabolomics. It facilitates expanding the chemical coverage of annotation and extending the assessment of metabolic pathways and activities, further revealing comprehensive chemical and metabolic insights towards biological processes.

## Results

**Unified AllCCS database**. To curate ion mobility CCS atlas, we develop the unified AllCCS database to store, standardize, and share the experimental and predicted CCS values. First, we collected 5119 reported experimental CCS values for 2193 compounds from 14 datasets, four laboratories, and two commercial IM–MS instruments (Supplementary Table 1). Then, we developed a five-step standardization procedure to clean up and unify all experimental CCS records, including collection of meta information, quality check, outlier removal, calculation of unified CCS values, and assignment of confidence levels ("Methods" and Supplementary Fig. 1). As a result, a total of 3539 unified CCS values with different adduct forms were calculated for 2193 compounds with definitive confidence levels (Fig. 2a). AllCCS provides wide-coverage of experimental CCS values for small molecules. Compared to other CCS databases, AllCCS is a platform to unify different CCS values, and overcomes the variations among different instruments and labs. The removal of outliers using trend line technique improved the accuracy, and has been validated in recent publications[30] (Supplementary Fig. 2). The unified CCS values are divided into level 1, level 2, level 3 and conflict with 462, 448, 2491, and 138 values, respectively (Supplementary Table 2). For example, 3,5-Diiodothyronine has a unified CCS value of 195.5 Å$^2$ for $[M + H]^+$ with confidence level 1, because it has been reported twice on drift tube IM-MS (DTIM-MS) from different labs and the maximum difference is within 1% (Supplementary Fig. 3). Currently, the unified CCS values comprises of 2423 cations and 1116 anions, and covered nine and six adducts in positive and negative modes, respectively (Fig. 2b and Supplementary Table 3). In terms of chemical diversity, they covered 15 super classes, 144 classes and 257 subclasses according to the definition of ClassyFire[34] (Fig. 2c and Supplementary Table 4). Among them, lipids and lipid-like molecules, organheterocyclic compounds, and benzenoids are the major super classes. We also compared the structural diversity of compounds in experimental AllCCS database with human metabolome database (HMDB) and DrugBank. The results showed that experimental CCS values covered 51.3% and 78.4% of chemical spaces of HMDB and DrugBank, respectively (Supplementary Fig. 4). These results demonstrated that compounds in AllCCS have a high diversity and representativeness of chemical structures. The unified CCS values are accessible in AllCCS webserver (http://allccs.zhulab.cn/).

**CCS prediction and performance benchmark**. In AllCCS, we further employed the new unified experimental CCS database, and optimized our machine-learning algorithm to predict CCS values of small molecules in a large-scale. Compared with MetCCS, AllCCS has several distinct features: (1) a large training dataset with high diversity of chemical structures (1873 compounds in total; Supplementary Data 1); (2) reduction of molecular descriptors to 15 and 9 for positive and negative modes, respectively; (3) development of representative structure

similarity (RSS) score to estimate prediction accuracy. The details for machine-learning-based prediction were provided in "Methods". Now, AllCCS includes a total of 1,670,596 compounds and 11,697,711 predicted CCS values, and covers seven popular compound databases—KEGG[35], HMDB[36], LMSD[37], MINE[38], DrugBank[39], DSSTox[40], and UNPD[41]. To the best of our knowledge, AllCCS is the largest and most comprehensive CCS database (Fig. 2d). All predicted CCS values were specified to confidence level 4, and have been deployed in AllCCS webserver. These records of compounds can be easily retrieved with different identifiers, such as SMILES, InChI, and InChIKey. AllCCS also supports users to predict CCS values for new compounds by inputting SMILES structures.

We validated the performance of AllCCS using two independent and external datasets, including validation set 1 (662 CCS values for metabolites and lipids; Supplementary Data 2), and validation set 2 (229 CCS values for drugs and natural products; Supplementary Data 3). Excellent consistencies between experimental and predicted CCS values were observed in both datasets (Fig. 2e, f and Supplementary Table 5). Specifically, for metabolites and lipids, the median relative errors (MREs) were 1.66% and 1.74%, while $R^2$ were 0.9901 and 0.9850 in positive and negative modes, respectively (Fig. 2e). For drugs and natural products, MREs were 1.81% and 2.25%, while $R^2$ were 0.9687 and 0.9230 in positive and negative modes, respectively (Fig. 2f). These results demonstrated AllCCS can predict CCS values with low errors for both endogenous and exogenous small molecules. Consistently, similar results were also obtained for different chemical classes and types of ion adducts (Fig. 2g and Supplementary Fig. 5). Taken together, AllCCS can accurately predict CCS values for small molecules with a vast diversity of chemical structures.

We also benchmarked the performance of AllCCS with MetCCS[27], DeepCCS[30], and ISiCLE[31] using validation sets (Fig. 2h–j and Supplementary Data 4). The results revealed that AllCCS made pivotal improvements of prediction accuracy. Specifically, there were 84% of CCS values with relative error <4% in AllCCS, and only 70%, 62%, and 28% for MetCCS, DeepCCS and ISiCLE for metabolites and lipids, respectively (Fig. 2h). Similar results were also observed for drugs and products, wherein 81%, 65%, 62%, and 35% of CCS values with relative error <4% were in AllCCS, MetCCS, DeepCCS, and ISiCLE, respectively (Fig. 2i). In addition to accuracy, AllCCS also has advantages in the prediction coverage and applicability. Specifically, AllCCS demonstrated the best prediction accuracy for most chemical super classes (8 out of 10), such as alkaloids and derivatives, benzenoids, and organic nitrogen compounds (Fig. 2j). For super classes such as alkaloids and derivatives and lipids and lipid-like molecules, only AllCCS can accurately predict CCS values with MRE <2%. Some examples were provided in Supplementary Fig. 6. Finally, AllCCS has made other seminal improvements compared with other tools, including implementation, time consuming and visualization (Supplementary Table 6). Collectively, these results demonstrated that AllCCS outperforms other tools in calculating CCS values in terms of accuracy and coverage.

**Structural similarity and CCS prediction accuracy**. We demonstrated that the structural similarity between the inputted chemical structures and the training dataset determines accuracy of predicted CCS values. To validate it, we divided our training and validation datasets into five super classes using ClassyFire in both positive and negative modes. We intentionally removed the compounds of one super class from the training dataset (for example, lipids and lipid-like molecules), and built machine-

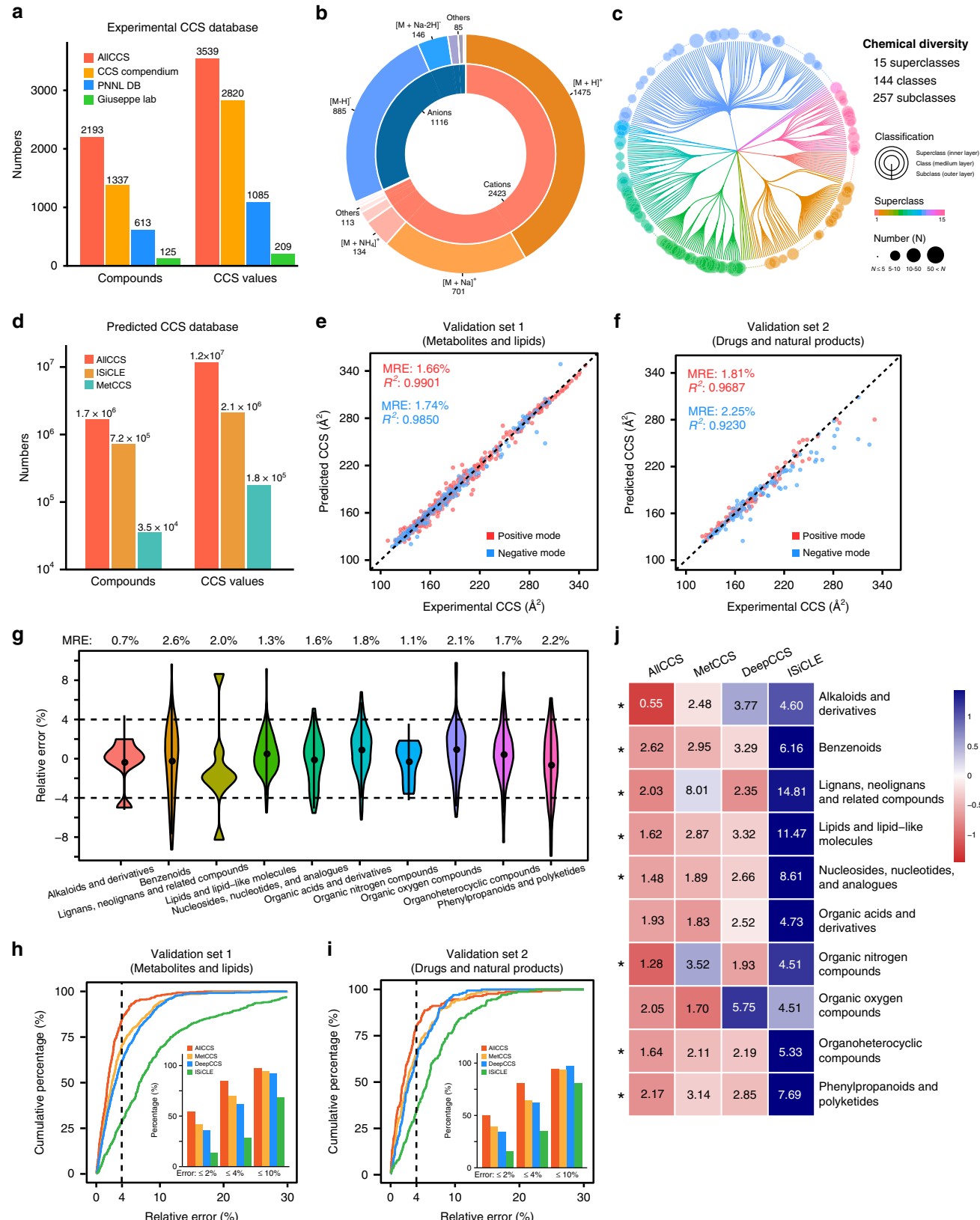

learning-based prediction model using the rest compounds as usual (Fig. 3a). Then, CCS values in validation sets were predicted and divided into two types: the excluded super class (i.e., lipids and lipid-like molecules) and other super classes, and further compared with the results in Fig. 2g. Taken lipids and lipid-like molecules as examples, the predicted CCS values in validation set showed significantly larger errors after excluding lipids from the training set (Fig. 3b). We also observed that other super classes have similar prediction errors between before and after excluding lipids from the training dataset. Then, we repeated the process for each super class, and observed similar results (Fig. 3c). The results demonstrated that the prediction errors of the excluded super

**Fig. 2 Unified AllCCS atlas for both experimental and predicted CCS values. a** Statistics of compounds and unified CCS values in AllCCS and other databases[12,24,26]; **b** statistics of adduct ions for unified CCS values in AllCCS; **c** chemical diversity for compounds with unified CCS in AllCCS, which was analyzed using ClassyFire; **d** statistics of compounds and predicted CCS values in AllCCS and other databases[27,31]; **e, f** correlations between predicted and experimental CCS values for external validation sets 1 (**e**) and 2 (**f**); **g** median relative errors (MREs) of predicted CCS values for 10 super classes of chemical structures; **h, i** cumulative percentages of predicted CCS values with indicated relative errors for external validation sets 1 (**h**) and 2 (**i**). The insert bar plot displayed the percentages of predicted CCS values within a certain relative error obtained from different tools; **j** heat map displaying the comparative prediction errors of ten super classes of chemical structures obtained from different tools; median relative error for each super class is shown in the pane while the color of pane is MRE normalized as Z-score. The symbol "*" represents that AllCCS has the lowest prediction error among these tools. Source data are provided as a Source Data file.

classes (MRE = 3.14%) were significantly larger than that of included super classes (MRE = 1.63%, Fig. 3d). Specifically, the prediction errors of CCS values in excluded super classes had an average relative error (ARE) as high as 9.16%. As a comparison, the included super classes only had an ARE of 2.20%. Therefore, the results confirmed that the structural similarity between the inputted chemical structures and the training dataset determines the CCS prediction accuracy.

**Representative structure similarity for accuracy evaluation**. In AllCCS, we further developed the representative structure similarity (RSS) score to quantify the structural similarity between the given compound and the training dataset, and to evaluate its relationship with the accuracy of predicted CCS values. RSS was calculated using molecular fingerprinting, and ranges from 0 to 1, representing completely different to highly similar structures compared to the training dataset (see "Methods"). Then, we found that there was a strong and significant correlation between RSS scores and the relative errors of predicted CCS values generated in the designed experiment in Fig. 3a (p-value = $3.19 \times 10^{-42}$; Fig. 3e). We empirically divided the RSS scores into small (RSS ≤ 0.6), medium (0.6 < RSS ≤ 0.8), and large (RSS > 0.8) groups. Large RSS group had significantly smaller MREs in CCS prediction than the small RSS group (1.56% vs. 5.90% in Fig. 3e). In AllCCS, we further validated the use of RSS score to evaluate CCS prediction accuracy in validation datasets (Fig. 3f). Compounds with small RSS had larger MRE (2.6%) than those with large RSS (1.7%). For examples, the CCS value of cardamonin was accurately predicted with a relative error 0.4% since it has a high RSS score of 0.9363. As a comparison, the CCS value of triclocarban had a large prediction error of 14.9% since it has a low RSS score of 0.5645 (Fig. 3g). Finally, we also calculated the RSS scores for compounds in common databases—KEGG, HMDB, LMSD, MINE, DrugBank, DSSTox, and UNPD, and found that >80% of compounds have high or medium RSS scores, and should be generated accurate CCS values (<2%) using AllCCS (Fig. 3h). Altogether, we proved that RSS implemented in AllCCS is able to evaluate and reflect the accuracy of CCS prediction, and AllCCS has a wide applicability for different small molecules.

**AllCCS improves known metabolite annotations**. With the large-scale AllCCS atlas, we first investigated its application for annotation of known compounds using the validation set 2 as examples. We matched the experimental m/z and CCS values of each compound to the whole AllCCS database (with ~1.67 million compounds). The average of candidates was significantly reduced from 1046 to 255 (76%) with the addition of CCS match (Fig. 4a and Supplementary Data 5). Similar results were obtained when performing multi-dimensional match using experimental m/z, as well as MS/MS and CCS values predicted by AllCCS and CFM-ID, respectively. The average of candidates was also significantly reduced from 553 to 144 (74%) with the addition of CCS match (Fig. 4a). We also demonstrated this using

experimental MS/MS spectral library from GNPS[42] with a total of 13,499 compounds. Similarly, the average candidates were reduced from 7.3 to 1.7 (77%) with the addition of CCS match (Supplementary Fig. 7). Therefore, ~75% of annotated candidates were filtered with the addition of CCS match. In addition, the candidate reduction through adding CCS match is effective with different database scales (Supplementary Fig. 8).

We also demonstrated that the addition of CCS match into the multi-dimensional match improved the rank of correct candidates (m/z + MS/MS + CCS vs. m/z + MS/MS matches; Fig. 4b). We found that the addition of CCS match improved the rank for most candidates ranging from 81.2% to 88.8% when different in-silico MS/MS tools (i.e., CFM-ID[43], MetFrag[44], and MS-FINDER[45]) were used (Fig. 4b). Similar results also were obtained in negative ionization mode (Supplementary Fig. 9). Taken the annotation of 6-hydroxycoumarin as an example, with the addition of CCS match, the number of potential candidates decreased from 956 to 181, and the rank for the correct candidate increased from 129th to 6th (Fig. 4c). Other annotation improvement examples were also provided in Supplementary Fig. 10.

Next, we demonstrated the annotation of known metabolites in biological samples with multi-dimensional match using m/z, CCS, and MS/MS spectra (Fig. 4d). Here, the KEGG and HMDB databases were used. For mouse embryonic fibroblast (MEF) cell sample, we putatively annotated a total of 2729 peaks from the acquired multi-dimensional liquid chromatography–ion mobility–mass spectrometry (LC–IM–MS/MS) data with multi-dimensional properties (Supplementary Fig. 11). Similarly, we validated that the average of candidates for these peaks was effectively reduced from 9.2 to 4.5 with the addition of CCS match (Fig. 4e, f). Similar results were also obtained for other biological samples, such as human plasma and fruit fly tissues (Fig. 4f and Supplementary Fig. 12). We found that 2038 out of 2729 peaks (75%) had reduced candidates with different degrees (Fig. 4g). Taken the feature M137T285C127 as an example, the annotated candidates were reduced step-wise from 9 to 2 with the integration of multi-dimensional match, and was finally annotated as hypoxanthine (1st rank; Fig. 4h). More examples were provided in Supplementary Fig. 13. In addition, we found the percentages of candidate reduction ranged 40–76% from high to low abundant features, indicating the better improvement for low abundant features (Supplementary Fig. 14). Combined, the integration of multi-dimensional properties especially the CCS match improved the annotation confidence with reduced false candidates and improved ranks.

**AllCCS enables unknown metabolite annotation**. Unknown metabolites are generated from uncharacterized resources, such as enzymatic transformation of endogenous metabolites, biotransformation of exogenous compounds (e.g., from environment) and gut microbiota. Here, we further investigated the integration of multi-dimensional properties (i.e., m/z + CCS + MS/MS) for unknown metabolite annotation in biological

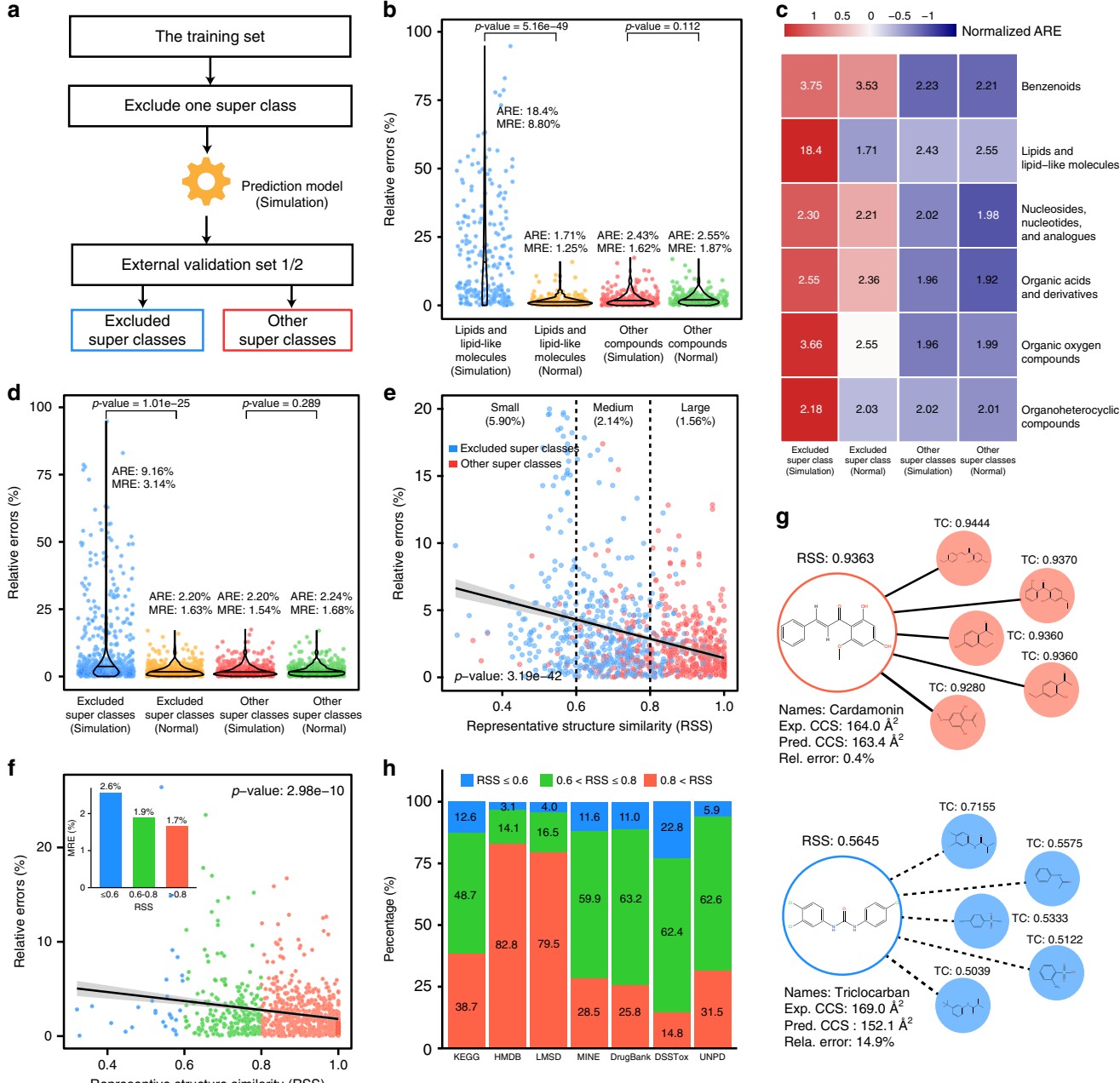

**Fig. 3 Representative structure similarity (RSS) and CCS prediction accuracy. a** The simulation workflow for investigating the structural similarity and prediction accuracy of CCS values; **b** comparison of CCS prediction accuracy between before and after excluding lipid and lipid-like molecules from the training set; the abbreviations "ARE" and "MRE" represent average relative error and median relative error, respectively; *p*-values determined by two-sided Wilcoxon rank-sum test; **c**, **d** comparison of CCS prediction accuracy between before and after excluding one super class from the training set; *p*-value determined by two-sided Wilcoxon rank-sum test; **e** correlation between the RSS scores and the relative errors of CCS prediction; the error bands represent 95% confidence interval; *p*-value determined by linear regression; **f** correlation of RSS scores and prediction errors in validation sets; the insert bar plot displays MREs for different RSS groups; the error bands represent 95% confidence interval; *p*-value determined by linear regression; **g** two compound examples for RSS and CCS prediction accuracy; The abbreviation "TC" represents tanimoto coefficient; **h** distribution of RSS scores in seven common compound databases. Source data are provided as a Source Data file.

samples (Fig. 5a). First, we generated the possible unknown metabolites from knowns in KEGG. We used all 16,023 metabolites in KEGG, and a total of 178 metabolic reactions and 117 enzymes to perform the in-silico enzymatic reaction[46] (Supplementary Data 6). Through this procedure, we have created a total of 100,404 possible unknown compounds via a 2-step in-silico enzymatic reaction, and expanded the chemical space of KEGG by sevenfolds (Fig. 5b). Among them, there are 5704 known unknowns (5.7%) for those included in PubChem but not in

KEGG, and 94,700 unknown unknowns (94.3%) for those not included both in either PubChem or KEGG (Supplementary Fig. 15). For example, the in-silico enzymatic reaction of phosphoenolpyruvate generated three unknown metabolites (Fig. 5c). For all generated unknown metabolites, we have calculated and predicted the *m/z* and CCS values using AllCCS and MS/MS spectra using MS-FINDER. This extended unknown database provides a potential for unknown metabolite annotation. With the multi-dimensional match to this database, we annotated both

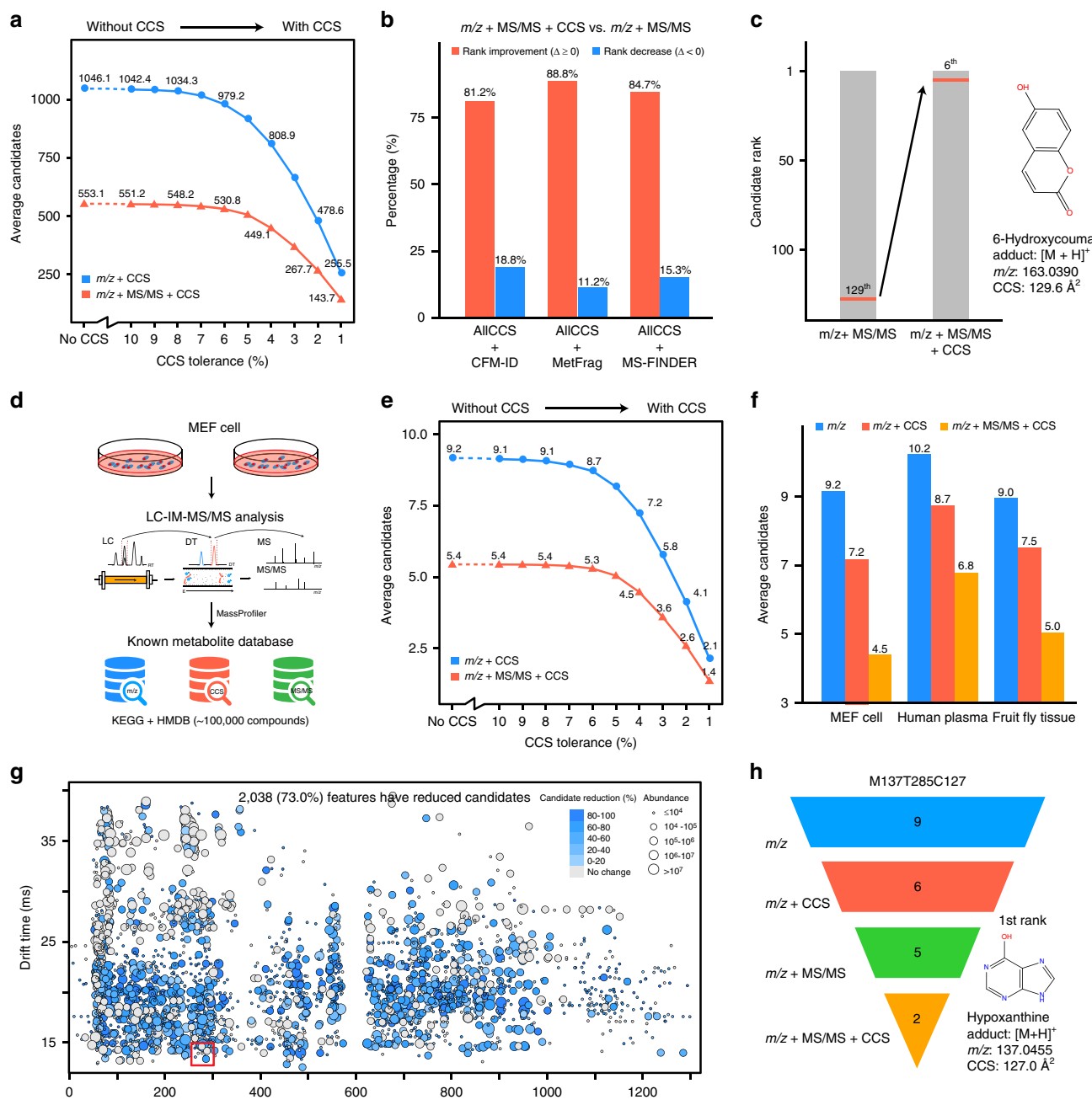

**Fig. 4 AllCCS improves known metabolite annotation. a** Candidate reduction with addition of CCS match demonstrated using validation set 2; **b** percentages of rank improvement for correct candidates with addition of CCS match to multi-dimensional match; three in-silico MS/MS prediction tools (CFM-ID, MetFrag, and MS-FINDER) were used to generate predicted MS/MS spectra; **c** an example of rank improvement for the compound 6-Hydroxycoumarin with addition of CCS match in multi-dimensional match; **d** schematic illustration of multi-dimensional annotation of metabolites in biological samples; **e** candidate reduction with addition of CCS match in metabolite annotation of MEF cell samples; **f** the candidate reductions in different biological samples; the CCS match tolerance was set as 4%; **g** percentages for candidate reduction with multi-dimensional match for each feature in MEF cell samples; **h** the hypoxanthine was successfully annotated with multi-dimensional properties; MS-FINDER was used to generate predicted MS/MS spectra for metabolite candidates. Source data are provided as a Source Data file.

known and unknown metabolites in mouse liver tissue samples. A total of 1223 features with 6092 metabolites were putatively annotated, including 2275 KEGG metabolites and 3817 unknowns (Fig. 5d and Supplementary Data 8). Among them, 67.0% of features had an unknown annotation. For example, the feature M384T767C189 (m/z: 384.1129; RT: 767s; CCS: 181.9 Å²) had 22 possible candidates in the extended database (Fig. 5e). Among them, 4 and 21 candidates were further reduced with

multi-dimensional match. Finally, an unknown metabolite (ExtDB016054) was annotated and further confirmed with the chemical standard (Fig. 5f and Supplementary Fig. 16).

We further investigated how AllCCS facilitated characterization of metabolic activities through unknown metabolite annotation. First, we analyzed 368 dysregulated metabolic features associated with aging in mice (p-value ≤ 0.05; 36-week vs. 104-week; Fig. 6a), and performed pathway enrichment analysis using

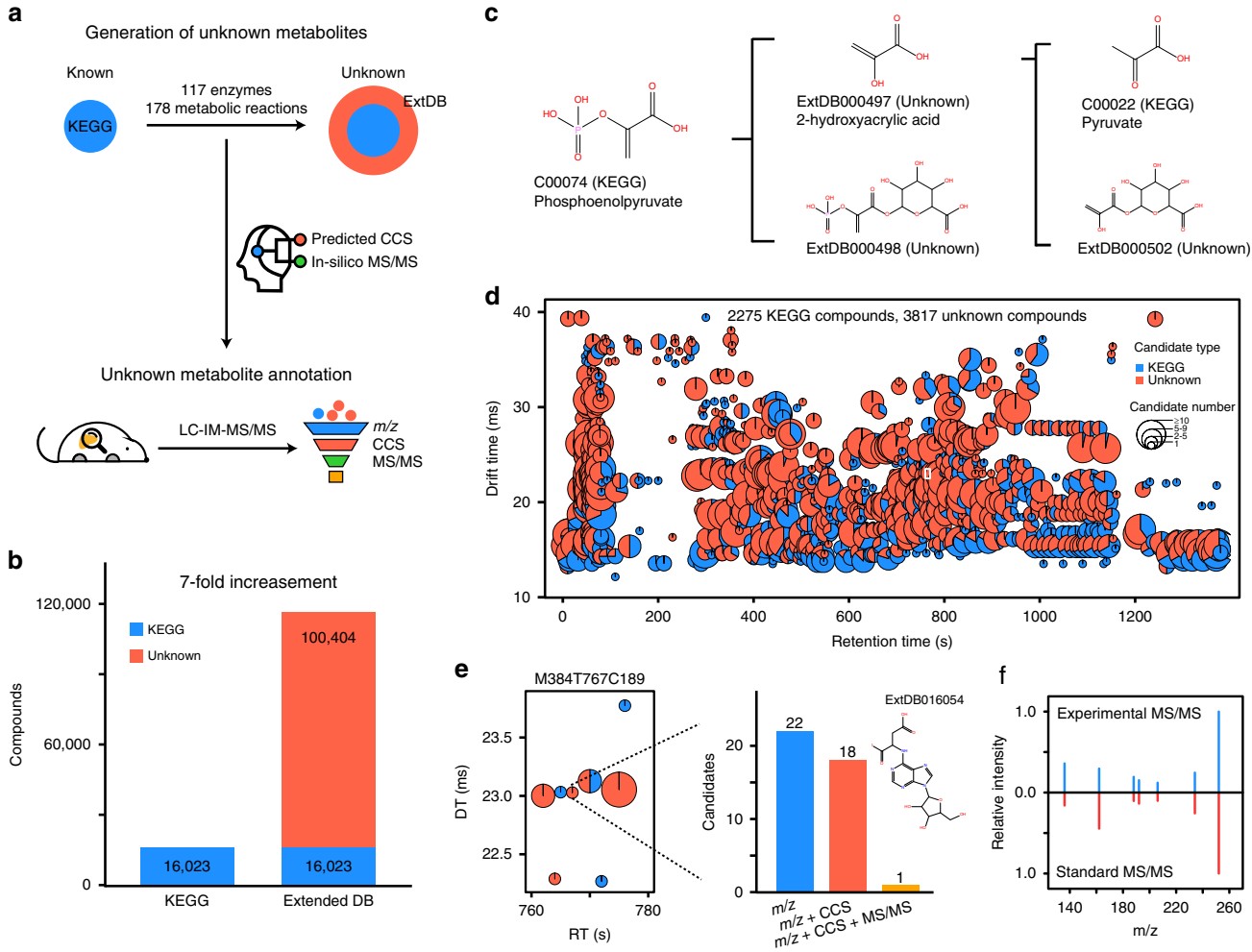

**Fig. 5 AllCCS enables unknown metabolite annotation. a** Schematic illustration of generation of possible unknown metabolites from KEGG and prediction of CCS and MS/MS spectra to support multi-dimensional match-based annotation; **b** number of unknown metabolites; **c** generation of unknown metabolites from phosphoenolpyruvate; **d** annotation of known and unknown metabolites in mouse liver with the multi-dimensional match; the inner pie plot is the composition of known and unknown metabolites; **e** an unknown metabolite annotation for the feature M384T767C189; **f** validation of unknown metabolite using chemical standard. Source data are provided as a Source Data file.

KEGG metabolites. Five metabolic pathways were enriched (p-value ≤ 0.05; Supplementary Fig. 17) and showed declined activities with aging (left panel in Fig. 6b), such as purine metabolism, nicotinate and nicotinamide metabolism, phenylalanine metabolism. The observations were consistent with previous reports[10]. Second, we extended the analysis from known KEGG metabolites to their related unknowns. Clearly, the unknown metabolites generated from five enriched metabolic pathways also had declined activities with aging (right panel in Fig. 6b). For example, adenylosuccinic acid and its derived unknown (ExtDB016054) were decreased during mouse aging (Fig. 6c). Interestingly, unknown metabolite ExtDB016054 showed more significant changes compared with its reactant precursor. Finally, we also performed chemical structure enrichment analysis for unknown metabolites, and 14 out of 174 subclasses were enriched (Supplementary Fig. 17). Among them, we found that several chemical subclasses related to purine metabolism showed declined activities with aging, such as pyrimidines and pyrimidine derivatives, and hydropyridines (Fig. 6d, e). Altogether, AllCCS facilitated expanding the metabolite coverage of annotation and extending the assessment of metabolic pathways and activities by providing new chemical structures.

## Discussion

In this work, we developed a large-scale ion mobility CCS atlas, namely, AllCCS, to support metabolite annotation in IM–MS-based metabolomics. Although several experimental CCS databases have been developed[22,24,26,47], AllCCS is unique because it provides a unified platform to store, standardize and share experimental CCS values from different labs and instruments. One possible concern is the inconsistency across different IM–MS instruments. A recent work from Schmitz group observed that most compounds have <1% errors between traveling wave IMS (TWIMS) and DTIMS, but some compounds showed larger deviations up to 6.2%[48]. Therefore, in AllCCS, we developed a five-step standardization procedure to automatically clean up and unify the experimental CCS records, which facilitates overcoming the variations from different instruments and labs. We believe the consistency of reported CCS values will be further improved with the launch of guidelines from IM–MS research community[17]. In addition, every user could access AllCCS webserver to view, calculate and download both experimental and predicted CCS values in AllCCS. We are also working on deploying CCS values into HMDB and other popular databases to make it available for the wider community. With the rapid growing of reported CCS

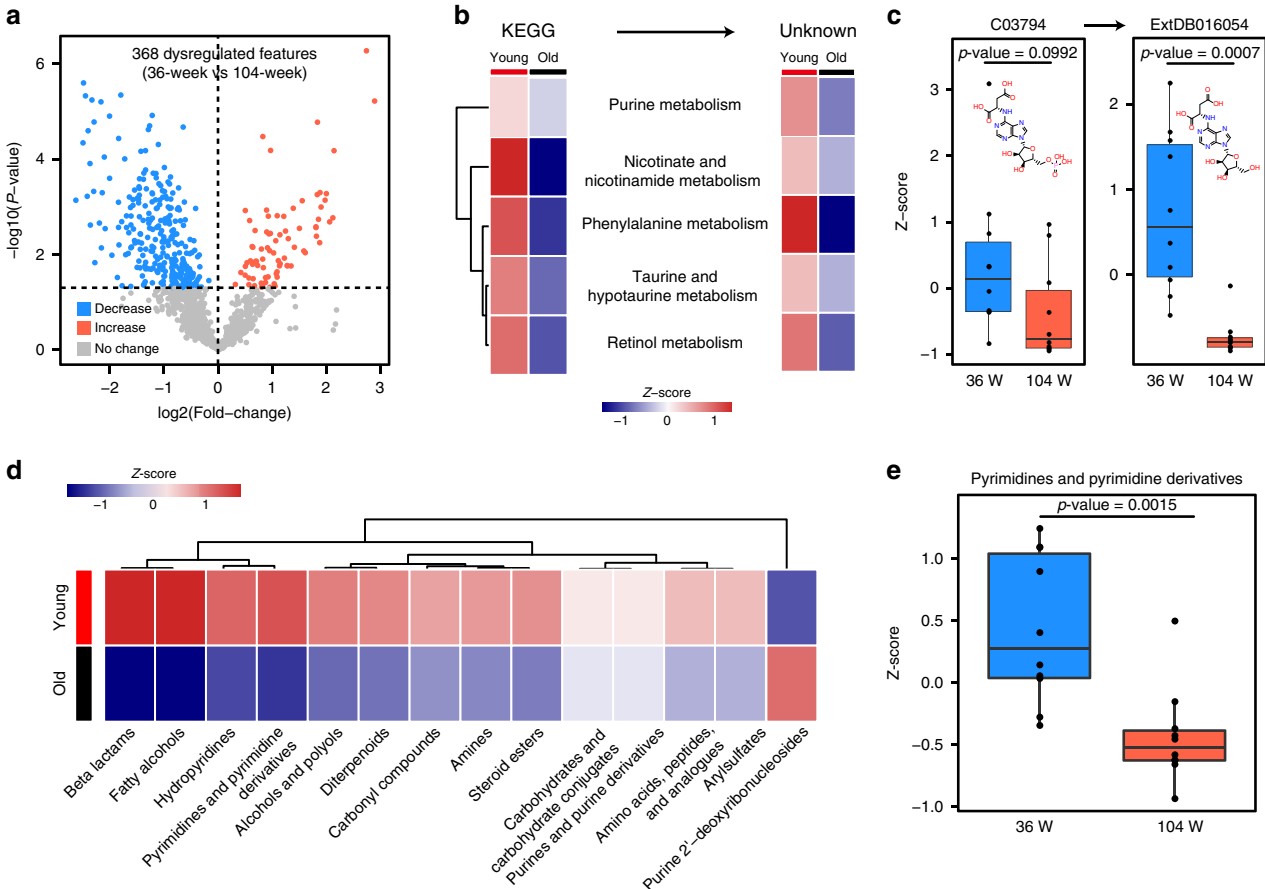

**Fig. 6 Expansion of metabolite coverage and assessment of metabolic activities with unknown annotation. a** Volcano plot showing the dysregulated features in aging mice (36-week vs. 104-week; *p*-value ≤ 0.05; two-sided Student's *t*-test); **b** heat maps showing age-dependent activities of five enriched KEGG metabolic pathways (left panel) and their related unknowns (right panel); **c** examples of adenylosuccinic acid and its derived unknown (ExtDB016054) down-regulated in aging mice (*n* = 10, biologically independent samples for each group; two-sided Student's *t*-test); **d** heat map showing age-dependent activities of 14 enriched chemical subclasses; **e** annotated unknowns from the subclass of pyrimidines and pyrimidine derivatives down-regulated in aging mice (*n* = 10, biologically independent samples for each group; two-sided Student's *t*-test). The lower, middle, and upper lines in box plots (**c**, **e**) correspond to 25th, 50th, and 75th quartiles, and the whiskers extend to the most extreme data point within 1.5 interquartile range (IQR). Source data are provided as a Source Data file.

values, AllCCS will be continually expanded and updated, making it as a valuable resource for both IM–MS and metabolomics research.

Different strategies have been developed for CCS calculation, such as MetCCS[27], LipidCCS[29], DeepCCS[30], ISiCLE[31], Dark-Chem[49], CCSbase[50] etc. However, the efficiency, accuracy and generalization capability for these methods need further improvements. Here, we demonstrated that AllCCS outperforms other tools in terms of efficiency, accuracy and coverage. The improvements of CCS prediction are attributed by three major factors (Supplementary Table 7): (1) AllCCS used large and wide-coverage CCS records to train the machine-learning-based prediction model; (2) the use of a five-step standardization strategy and unified CCS values overcome the biases across different instruments and labs and improved the data quality of the training set; (3) the selection of optimized molecular descriptors (MDs) improved the prediction accuracy. In addition, AllCCS includes the representative structure similarity (RSS) score to estimate the prediction accuracy for one compound. However, some limitations in CCS prediction are still presented. For example, most of metabolite ions have their unique CCS values, but some may have multiple CCS values for different conformations. Currently, AllCCS and other tools can only predict one CCS value for one conformation, presumably the most

compact one. The introduction of quantum chemistry for conformation generation and 3-D molecular descriptors into the machine-learning-based prediction may help to address this challenge. The second challenge is the CCS prediction of isomers (e.g., cis-trans isomers in lipids). Although the CCS prediction has made effective improvements to ~2% prediction errors, identification of metabolite isomers is still a challenge due to the limit resolution of ion mobility separation (e.g., 40–60 for DTIMS and TWIMS). We demonstrated this challenge with an example of four monosaccharide phosphate isomers, which were poorly separated with IM (Supplementary Fig. 18). However, these isomers would be partially separated with an IM resolution of 200, and baseline separated with an IM resolution of 500. Therefore, both CCS prediction and annotation accuracy will be further improved with the availability of high-resolution IM instruments, such as TIMS and cyclic IM. Finally, although we focus on metabolites, AllCCS also supports the CCS prediction for other small molecules, like drugs, natural products, pesticides etc.

Metabolite annotation is one of the major bottlenecks for untargeted metabolomics. Metabolite annotation usually requires high quality of experimental MS/MS spectra. However, many experimental factors, such as high sample complexity, low concentration and co-elution of isobaric metabolites, present challenges to acquire high quality of MS2 spectra. In contrast, the

measurement of CCS value is less affected by experimental factors, and could be accurately acquired from molecular ions even with low abundances. Now, the use of multi-dimensional match, including CCS match provides multi-dimensional characterization of metabolites. For example, ~75% of low abundant features have reduced candidate numbers with the addition of CCS match (Supplementary Fig. 14). We demonstrated that the addition of CCS values to the multi-dimensional match improved the annotation confidence with reduced false candidates and improved ranks of correct candidates. Currently, AllCCS does not directly process raw IM–MS data files. Instead, we aim to incorporate AllCCS into other data processing tools (e.g., MS-DIAL4[51]) to accelerate the workflow from raw data processing to compound identification. In addition, developers could also integrate AllCCS with other data processing software tools such as XCMS[52] and MZmine[53].

Unknown metabolites are generated from endogenous and exogenous resources, and have no standard MS/MS spectra available. Currently, unknown metabolite annotation is mainly performed using in-silico MS/MS tools (e.g., MS-FINDER[45] and SIRIUS[54]). In this work, we further demonstrated that AllCCS provided a promising strategy to integrate the predicted CCS and in-silico MS/MS tools for unknown annotation, and facilitated extending the assessment of metabolic pathways and activities with new chemical structures. These unknown metabolites were created from KEGG compounds via in-silico enzymatic reactions, because KEGG covers ~6000 species and various compounds, including primary metabolites, secondary metabolites from plant and bacterium, and xenobiotic compounds[55]. In the future, other databases such as RECON[56] are also good choices for metabolic reconstruction towards human metabolism study. Finally, we believe this strategy will have more powerful prospects when combined with other data processing tools (e.g., MS-DIAL4) and advanced algorithms (e.g., GNPS[9] and MetDNA[10]). Taken together, the AllCCS atlas has provided a high-quality and unified CCS database for IM–MS, and further opens a new avenue for known and unknown metabolite annotation in IM–MS-based untargeted metabolomics.

## Methods
**Curation of the unified CCS database**. A total of 5119 experimental CCS values were collected from 14 datasets, 4 independent labs, and 2 instrument platforms (Supplementary Table 1), which were reported in recent publications from 2015–2018. To curate the unified CCS database, each dataset was cleaned and standardized with a five-step procedure as follows (Supplementary Fig. 1).

(1) Collection of meta information. For each CCS record, the chemical translation service[57] [http://cts.fiehnlab.ucdavis.edu/] was utilized to generate the chemical identifiers for compounds, such as InChIKey, CAS number, PubChem CID, etc. Then, the SMILES structure for each compound was generated using an R package *rinchi* [https://github.com/CDK-R/rinchi]. The compound formula and exact masses for different adducts (Supplementary Table 8) were calculated using an R package *rcdk* [https://cran.r-project.org/web/packages/rcdk/index.html]. Finally, the chemical classification for each compound was obtained using ClassyFire[34] [http://classyfire.wishartlab.com/].

(2) Quality check. Some CCS records were intentionally removed for those without chemical structures, with ion adducts not included in Supplementary Table 8, or having large *m/z* errors (>10 ppm). Then, for each dataset, we also removed the inconsistent CCS records from the same instrument platform. For one ion adduct with more than one CCS record, the maximum differences between CCS records were calculated. If the maximum difference was >0.5%, the related CCS records were removed. Otherwise, the averaged CCS value was calculated and assigned as the CCS record.

(3) Outlier removal. The CCS outliers were further removed using the CCS trend lines, which was similar to the CCS compendium[26]. The trend line of each super class ($n \geq 10$) was fitted by a power function, and the CCS records exceeding 99% of the predictive interval were removed. A total of 103 CCS outliers were removed, and two examples of outliers were confirmed in a recent publication[30] (Supplementary Fig. 2).

(4) Calculation of unified CCS values. The CCS values from different instrument platforms were further merged as unified CCS values. The unified CCS value is an average of CCS values from different instrument platforms, which is specific to the compound and its adduct. Specifically, for one ion adduct, if it had multiple CCS records obtained from DTIM-MS, the unified CCS value was the average value from the CCS records in DTIM-MS. Otherwise, the unified CCS value was calculated using all CCS records from different platforms. A total of 3539 unified CCS values were generated.

(5) Assignment of confidence levels. For each unified CCS value, we assigned a confidence level using the following rules: Level 1: the unified CCS is calculated using experimental CCS records from ≥2 independent datasets in DTIM-MS instruments, and the maximum CCS difference is ≤1%; Level 2: the unified CCS is calculated using experimental CCS records from ≥2 independent datasets in different commercial instruments (DTIM-MS, TWIM-MS, or TIMS-MS), and the maximum CCS difference is ≤3%; Level 3: the unified CCS is only reported in one dataset from commercial instruments (DTIM-MS, TWIM-MS, or TIMS-MS); Conflict: the unified CCS is calculated using experimental CCS records from ≥2 independent datasets in different commercial instruments (DTIM-MS, TWIM-MS, or TIMS-MS), but the maximum CCS difference is >3%. All predicted CCS values were assigned as level 4 in AllCCS.

**Training and validation sets for CCS prediction**. AllCCS employed the unified CCS values for CCS prediction and validation. Specially, 80% of unified CCS values (1851 and 795 CCS values in positive and negative modes, respectively) were randomly selected as the training set (Supplementary Data 1). Here, we only kept seven most common adducts ($[M + H]^+$, $[M + Na]^+$, $[M + NH_4]^+$ and $[M + H-H_2O]^+$ for positive mode; $[M-H]^-$, $[M + Na-2H]^-$, $[M + HCOO]^-$ for negative mode), and removed CCS values with the confidence level of conflict. In addition, two datasets were used for performance validation: (1) external validation set 1 (metabolites and lipids) consists of 463 and 199 CCS values in positive and negative modes, respectively (Supplementary Data 2); (2) external validation set 2 (drugs and natural products) consists of 107 and 122 CCS values in positive and negative modes, respectively (Supplementary Data 3). Both validation sets were acquired using chemical standards on Agilent DTIM-MS 6560. The acquisition of CCS values and the standard MS/MS spectra followed the previous publications[27].

**Molecular descriptor calculation and selection**. For each compound, a total of 221 molecular descriptors (MDs) were calculated using the SMILES structure and the R package rcdk. Among them, non-differential MDs were first removed. The missing values for the rest MDs were imputed using the KNN algorithm. All MD values were normalized to *Z*-score and subjected to selection using the recursive feature elimination with cross validation (RFECV) algorithm (Supplementary Fig. 19). In order to eliminate the scale effect of the training set, 50%, 60%, 70%, 80%, or 90% of the training set were used for RFECV. For each condition, the RFECV was performed by 200 times (1000 times in total). In each RFECV, the least important MD was recursively removed according to the coefficient of the LASSO regression via a tenfold cross validation. The MD combination with highest scores in the cross validation were kept. Finally, MDs with the frequency >700 in 1000 RFECV replications were ultimately selected. In positive and negative modes, 15 and 9 MDs were selected, respectively (Supplementary Table 9). We also demonstrated that the selected MDs showed smaller prediction errors than those obtained from the step-wise selection or the random selection (Supplementary Fig. 20 and Supplementary Table 10). The python software sklearn [https://scikit-learn.org/stable/] was used for RFECV.

**Support vector regression-based CCS prediction**. The support vector regression (SVR) algorithm was used to develop the CCS prediction using the selected MDs and CCS values in the training set. The general workflow was similar as our previous publications[29]. Briefly, two hyper-parameter cost of constraints violation (*C*) and gamma (*γ*) were optimized from 105 combinations via a tenfold cross validation with 100 repeats. Seven groups of *C* value (0.001, 0.005, 0.025, 0.05, 0.1, 0.25, 0.5)/$N_{MD}$ and 15 groups *γ*-value (2 to $2^{15}$) were set for parameter optimization. Radial basis function was employed for kernel function. $N_{MD}$ represented the number of selected MDs. Finally, the hyper-parameter combinations were selected as follows: *C*, 0.1/15 and 0.1/9 in positive and negative modes, respectively; *γ*, $2^8$ and $2^{13}$ in positive and negative modes, respectively. As a result, 1.67% and 1.72% of MREs were obtained for the training set in positive and negative modes, respectively (Supplementary Table 11). In addition, the high gamma parameters indicated that the optimized parameters in SVR prediction make the model towards a linear regression, but has better performances comparing to multiple linear regression (Supplementary Table 12).

**Representative structure similarity**. The representative structure similarity (RSS) was calculated to characterize the structure similarity between the inputted structure and the training set (Supplementary Fig. 21). The molecular fingerprint of inputted structure was first computed using the R package *rcdk*. Then, the structure similarity between the inputted structure and each structure in the training set was

calculated using the tanimoto coefficient (TC) shown as follows:

$$TC_{(StrA,StrB)} = \frac{N_{StrA \cap StrB}}{N_{StrA} + N_{StrB} - N_{StrA \cap StrB}} \quad (1)$$

where $N_{StrA}$ and $N_{StrB}$ were the molecule fingerprints of structures A and B, respectively, and $TC_{(StrA,StrB)}$ was the TC between structure A and structure B. Here, structure A was the inputted structure and structure B was a structure in training set. $N_{StrA \cap StrB}$ was the intersection set of structure A and B. Then, RSS score of the inputted structure was calculated using the average of top five TCs:

$$RSS_{StrA} = \sum_{i=1}^{5} TC_i / 5 \quad (2)$$

where $RSS_{StrA}$ was the RSS of the inputted structure A, and $TC_i$ represented top $i$ tanimoto coefficient.

**Benchmark of CCS prediction performance**. The generation of CCS values using MetCCS[27], DeepCCS[30], and ISiCLE[31] for compounds in the external validation sets was performed as follows. For MetCCS, the webserver [http://www.zhulab.cn/MetCCS/] was used to predict CCS values. The inputted molecular descriptors of each compound were calculated by ChemAxon MarvinSketch (Version 16.10.24) and ALOGPS [http://www.vcclab.org/web/alogps/]. For DeepCCS, CCS values were calculated using the SMILES structures and the python package downloaded from the internet (https://github.com/plpla/DeepCCS, on April 2nd, 2019). For ISiCLE, CCS values generated from ISiCLE Lite v0.1.0 were directly downloaded from the webserver [https://metabolomics.pnnl.gov/ccs/] on March 11th, 2019. All CCS values were provided in Supplementary Data 4.

**AllCCS webserver**. The AllCCS webserver was hosted on a Linux server from Alibaba Cloud, and free-accessible for non-commercial use via http://allccs.zhulab.cn/. AllCCS webserver has three major functions: (1) the unified and predicted CCS databases, (2) the CCS prediction, and (3) metabolite annotation. The predicted AllCCS database contains a total of 1,670,596 compounds and 11,697,711 predicted CCS values. These compounds are downloaded from KEGG[35], HMDB[36], LMSD[37], MINE[38], DrugBank[39], DSSTox[40], and UNPD[41] databases (Supplementary Table 13). The CCS prediction function provides a visualized interface for users to predict CCS values with the inputted SMILES structures. The metabolite annotation provides a feature match function to search the AllCCS database with experimental $m/z$ and CCS values. In addition, it also provides a candidate rank function to perform multi-dimensional annotation by integrating the annotation results from in-silico MS/MS prediction tools. The tutorial of AllCCS is available on the website.

**CCS match, MS/MS match, and multi-dimensional match**. A trapezoidal score function was developed to measure the CCS match. First, it removed the candidates with CCS values exceeding the maximum tolerance, then calculated the CCS match score ($S_{ccs}$) using a trapezoidal function as Eq. 3:

$$S_{ccs} = \begin{cases} 1, & \Delta_{rela} \, TOL_{min} \\ 1 - \frac{(\Delta_{rela} - TOL_{min})}{TOL_{max} - TOL_{min}}, & TOL_{min} \le \Delta_{rela} \le TOL_{max} \\ 0, & \Delta_{rela} > TOL_{max} \end{cases} \quad (3)$$

where $TOL_{min}$ and $TOL_{max}$ are minimum and maximum tolerances, respectively. The default values for $TOL_{min}$ and $TOL_{max}$ are 2% and 4%, respectively. The $\Delta_{rela}$ is relative CCS error calculated as Eq. 4.

$$\Delta_{rela} = \frac{\left| CCS_{Pred} - CCS_{Exp} \right|}{CCS_{Exp}} \times 100 \quad (4)$$

The experimental MS/MS spectra and their possible candidates were imported into in-silico MS/MS tools to perform MS/MS match. Three in-silico MS/MS prediction tools, such as MetFrag[44], CFM-ID[43], and MS-FINDER[45] were used in this work. The format of imported data was modified according to the requirements of each tool. The brief procedures are described as follows: (1) MetFrag: the command line version MetFragCL (version 2.4.5-CL) was downloaded from https://ipb-halle.github.io/MetFrag/, and the parameter file was generated via R package ReSOLUTION [https://github.com/schymane/ReSOLUTION]; (2) CFM-ID: the software version 2.4 was downloaded from https://sourceforge.net/projects/cfm-id/files/. The pre-trained model params_se_cfm and the parameter file param_output0.log were used. The predicted MS/MS spectra were provided as MSP format in Supplementary Data 5. (3) MS-FINDER: the software version 3.24 was downloaded from http://prime.psc.riken.jp/Metabolomics_Software/MS-FINDER/index.html, and run with the console. The detail parameters of each tool were provided in Supplementary Table 14. The experimental MS/MS spectral library was downloaded from GNPS with a total of 13,499 compounds (https://gnps.ucsd.edu/ProteoSAFe/libraries.jsp; accessed on May 23th, 2020). The spectral match utilized reverse dot-product scores, and its parameters were kept same with our previous publication[32].

Multi-dimensional match was performed by integrating the CCS match score and MS/MS match score as Eq. 5:

$$S_{integrated} = W_{ccs} \times S_{ccs} + W_{MS/MS} \times S_{MS/MS} \quad (5)$$

where $S_{CCS}$ and $S_{MS/MS}$ are CCS and MS/MS match scores, respectively. Here, $S_{MS/MS}$ is the similarity between experimental MS/MS and in-silico MS/MS, which is obtained from different in-silico MS/MS tools with different scoring methods. The $S_{MS/MS}$ is rescaled to 0–1 before integration. The $W_{ccs}$ and $W_{MS/MS}$ are weights for the CCS and MS/MS match scores, respectively. The $W_{ccs}$ and $W_{MS/MS}$ were optimized as 0.3 and 0.7, respectively (Supplementary Fig. 22).

**Chemicals**. LC–MS grade methanol (MeOH) and water ($H_2O$) were purchased from Honeywell (Muskegon, MI, USA). LC–MS grade acetonitrile (ACN) was purchased from Merck (Darmstadt, Germany). LC–MS grade methylene dichloride ($CH_2Cl_2$) was purchased from Fisher Scientific (Morris Plains, NJ, USA). Ammonium hydroxide ($NH_4OH$) and ammonium acetate ($NH_4OAc$) were purchased from Sigma (St. Louis, MO, USA). The chemical standard succinoadenosine was purchased from J&K (Shanghai, China), while other chemical standards were purchased from TopScience (Shanghai, China).

**Sample preparation**. Aging mouse liver tissues (c57BL-6J; 36-week vs. 104-week; $n = 10$ for each group) were dissected, frozen with liquid nitrogen, and stored at $-80\,°C$. The mouse tissue studies were approved by Animal Ethics and Welfare Management Committee of Interdisciplinary Research Center on Biology and Chemistry, Chinese Academy of Sciences (Shanghai, China). Metabolite extraction followed our published protocol[10]. In brief, 10 mg of mouse liver tissues was firstly homogenized with 200 μL of $H_2O$ and 20 ceramic beads (diameter, 0.1 mm) using a homogenizer (Precellys 24, Bertin Technologies) at the low-temperature condition. The protein concentration of the homogenized solution was measured with the Pierce BCA Protein Assay Kit (Catalog No. 23225, Thermo Fisher) for normalization. One-hundred microliters of homogenized solution was used for metabolite extraction. A total of 100 μL of $H_2O$ and 800 μL of solvent mixture of ACN:MeOH (1:1, v/v) was added, and vortexed for 30 s, and sonicated for 10 min at 4 °C water bath. After incubation for 1 h at $-20\,°C$, the sample was further centrifuged for 15 min at $16,200 \times g$ and 4 °C. The supernatant was collected and evaporated to dryness at 4 °C. The dry extracts were then reconstituted into 100 μL of ACN:$H_2O$ (1:1, v/v), followed by sonication at 4 °C for 10 min, and centrifuged at $16,200 \times g$ and 4 °C for 5 min to remove the insoluble debris before LC–IM–MS/MS analysis.

Other biological samples were prepared as follows. For plasma, 100 μL of human plasma (Catalog No. HPH-0500, Equitech-Bio. Inc, USA) was extracted using 400 μL of solvent mixture of MeOH:ACN (1:1, v/v) in the centrifuge tube, and then the mixture was vortexed for 30 s and sonicated for 10 min at 4 °C water bath. The rest of the procedure was the same as described for mouse liver tissue sample. For cell samples, RIPK1$^{-/-}$ mouse embryonic fibroblasts (MEFs) cell line (generated from RIPK1 KO mice) were provided from Prof. Junying Yuan's Lab (Chinese Academy of Sciences, Shanghai). One milliliter of MeOH:ACN:$H_2O$ (2:2:1, v/v/v) solvent mixture was added to the samples, followed by vortex for 30 s and sonication for 10 min at 4 °C water bath. Then the samples were incubated in liquid nitrogen for 1 min, thawed on ice, and sonicated for 10 min at 4 °C water bath. The above vortex–freeze–thaw cycle was repeated three times. The rest of the procedure was the same as described for mouse liver tissue sample. For fruit fly head samples, the sample collection and extraction followed our previous publication[10].

**LC–IM–MS/MS analysis**. A UHPLC system (Agilent 1290 series) coupled to a quadruple time-of-flight mass spectrometer equipped with an ion mobility drift tube (Agilent DTIM-QTOF-MS 6560, Agilent Technologies, USA) was used for LC–IM–MS/MS data acquisition. The LC separation was performed on a Waters BEH Amide column (particle size, 1.7 μm; 100 mm (length) × 2.1 mm (i.d.)) maintained at 25 °C. The solvent A was 100% $H_2O$ with 25 mM $NH_4OAc$ and 25 mM $NH_4OH$, and solvent B was 100% ACN. The flow rate was 0.3 mL/min, and the gradient was described as follows: 0–1 min: 95% B, 1–14 min: 95% B to 65% B, 14–16 min: 65% B to 40% B, 16–18 min: 40% B, 18–18.1 min: 40% B to 95% B and 18.1–23 min: 95% B. The sample injection volume was 2 μL.

The data acquisition was operated in IM-Q-TOF mode. The source parameters were set as follows: sheath gas temperature, 325 or 275 °C in positive or negative modes; dry gas temperature, 300 °C; sheath gas flow, 11 L/min; dry gas flow, 8 L/min; capillary voltage, 4000 V or $-3000$ V in positive or negative modes, respectively; and nebulizer pressure, 20 or 25 psi in positive or negative modes, respectively. The TOF mass range was set as $m/z$ 50–1700 Da. For ion mobility parameters, the nitrogen ($N_2$) was used for the drift gas. Other related IM parameters were set as follows: entrance and exit voltages of drift tube, 1600 and 250 V; trap filling and trap release times, 20,000 and 150 μs. The pressure of drift tube was set at 3.95 Torr. The MS/MS spectra were acquired in the "Alternating frames" mode, and the collision energy was fixed at 20 V in frame 2. The CCS values were calculated with single electric field method. All data acquisitions were carried out using MassHunter Workstation Data Acquisition Software (Version B.08.00, Agilent Technologies, USA).

Chemical standards were first dissolved at 0.01 mg/mL in either $H_2O$, MeOH, $CH_2Cl_2$, DMSO, or their mixture with different proportions depending on compound polarity and solubility, and subject to measurements of CCS values and MS/MS spectra. The CCS values were independently measured three times across 2 months using a single-field approach on Agilent DTIM-QTOF-MS 6560 instrument according to our previous publication[27]. The MS/MS spectra were acquired using targeted MS/MS method with three different collision energy levels (10, 20, and 40 V).

**Data processing and metabolite annotation**. Raw MS data files (.d) were first recalibrated using IM–MS Reprocessor (Version B.08.00, Agilent Technologies). Then, the smoothing and saturation repair were performed using PNNL Pre-Processor (Version 2018.06.02). The CCS calibration was performed by IM–MS Browser software (Version B.08.01, Agilent Technologies). The pre-processed data files were submitted for feature finding, alignment, and MS/MS spectra extraction using Mass Profiler (Version 10.0, Agilent Technologies). Finally, the peak table and MS/MS spectra (CEF format) files were exported for metabolite annotation. One MS/MS spectrum with highest intensity was selected for each feature, similar to the protocol in LipidIMMS Analyzer[32]. The detail parameters of data processing tools were provided in Supplementary Table 15. The metabolites were annotated using multi-dimensional match as we described before. The $m/z$ tolerance was set at 25 ppm, and only $[M+H]^+$ and $[M-H]^-$ adducts were considered for positive and negative modes, respectively. The MS-FINDER was used for in-silico MS/MS match, and kept chemical structures within top 3 formulas for unknown metabolite annotation. The known metabolite database (KEGG and HMDB) and the extended database were used for known and unknown metabolite annotation, respectively.

**Generation of unknown metabolites**. Unknown metabolites were generated based on in-silico enzymatic reaction via BioTransformer[46] (version 1.0.8). The command line tool was used and downloaded from [https://bitbucket.org/djoumbou/biotransformer/src/master/]. The SMILES structures of KEGG compounds were used for in-silico reaction, and the "EC-based transformation" was used for metabolic transformation. The reaction step was set as 2. All generated metabolites were merged by InChIKey, and their SMILES structures were converted via Open Babel[58]. Finally, a total of 100,404 unknowns were finally generated and included in the extended database (Supplementary Data 6). These compounds and their predicted CCS values were also provided in AllCCS webserver.

**Metabolic pathway and structure enrichment analysis**. For the analysis of aging mice samples, the peak intensity table from Mass Profiler was first normalized to the protein concentration from BCA. Then, zero imputation with KNN algorithm was performed. Student's $t$-test was used for calculating $p$-value. The metabolic pathway and chemical structure enrichment analyses were performed via hypergeometric test[59] and Kolmogorov–Smirnov (KS) test[60], respectively. All chemical classes of unknowns were obtained using ClassyFire. The quantitative analysis followed our previous publication[10], and the $z$-scale normalization of peak intensities was used in this work.

**Reporting summary**. Further information on research design is available in the Nature Research Reporting Summary linked to this article.

## Data availability
All raw data files can be accessed at MetaboLights (MTBLS1622 and MTBLS1693). The annotation results for all metabolomics datasets were provided in the Supplementary Data 7 and 8. The unified CCS database can be accessed in AllCCS webserver with free registration. Source data are provided with this paper.

## Code availability
The source code of AllCCS prediction was provided in Github [https://github.com/ZhuMetLab/AllCCS]. All functions (database search, CCS prediction and annotation) are also provided in the AllCCS webserver [http://allccs.zhulab.cn/] via a free account.

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

## Acknowledgements

The work was supported by National Key R&D Program of China (2018YFA0800902), National Natural Science Foundation of China (31971356), Shanghai Municipal Science and Technology Major Project (2019SHZDZX02), and Chinese Academy of Sciences Major Facility-based Open Research Program.

## Author contributions

Z.J.Z. and Z.Z. conceived the idea and designed the database and software. Z.Z. developed the AllCCS atlas and prediction program. M.L., X.C., R.W., and Z.Z. performed the sample preparation, data acquisition and data processing. X.C. and Z.Z. contributed to the tutorial of AllCCS webserver. M.L. and Z.Z. performed the data analysis. Y.Y. deployed the AllCCS webserver. Z.J.Z., Z.Z., M.L. and X.C. tested and debugged the program and webserver. X.X. contributed to part of codes in molecular descriptor selection. Z.J.Z. and Z.Z. wrote the manuscript. Z.J.Z. supervised the project.

## Competing interests

The authors declare no competing interests.
