## [Peer Review File · Nature Communications]

REVIEWER COMMENTS

Reviewer #1 (Remarks to the Author):

This paper described the creation and validation of an atlas of CCS values. The atlas is created using a combination of experimental CCS (merged from multiple databases) and in-silico predicted CCS. The authors show that they are able to predict CCS better than current systems, and that combining predicted CCS with other data (m/z, predicted MS/MS) improved candidate ranking. The atlas looks to be a resource that has practical utility and the authors do a good job showing that it performs well.

1. Overall I enjoyed reading this paper. I think it does a good job in setting out the system, evaluating it, and describing the benefits it brings. There are quite a few places in the text though in which more thorough proof-reading is required (too numerous to list here).

2. The improvement in CCS prediction is good, and the ML methodology looks sound. I would welcome a little more discussion on where the authors think this improvement comes from. Is it simply the larger unified database?

3. The SVM regression for CCS prediction: the gamma parameters (note: radial basis, not radio basis) seem extraordinarily high (2^8 and 2^{13}). Depending on the way the RBF function is parameterised, this effectively becomes either a linear regression, or something akin to a nearest neighbours scheme. Not necessarily a problem, but something that if I saw it in an analysis, would ring some alarm bells for me. Have the authors looked into this?

4. When combining with MS/MS, the authors use only in-silico MS/MS on their side. Why is this? in-silico MS/MS prediction is still not great. For the "unknown" experiments, predicted MS/MS is obviously the only route. However, could the authors not have done some experiments with real MS/MS library spectra? Playing devils advocate: perhaps the improvement CCS gives about in-silico MS/MS is because in-silico MS/MS is not reliable. Would we see the same improvements with real MS/MS libraries? There are plenty in the public domain (Massbank, GNPS, HMDB, etc).

5. Reproducibility: the authors do a good job of providing the supplementary material to allow another to produce their results. One thing that I cannot find are the in-silico predicted spectra that were used in the experimental section. Recreating these would be challenging (e.g. CFM-ID takes a

long time to run) and could prohibit reproduction. I think they should be included as, e.g., one or more .mgf files.

Reviewer #2 (Remarks to the Author):

The manuscript by Zhou and colleagues describes a software tool for the large scale deployment of ion mobility collision cross-section data for metabolite identification of both known and unknown metabolites. While a number of tools have been created previously, including by the senior author, to search databases of known standards and also predict CCS values for metabolites without CCS measures, AllCCS dramatically extends the use of CCS values and ion mobility for metabolite identification and annotation by unifying a number of experimental CCS values to produce consensus values and predicting CCS values for an impressive database of putative metabolites. The authors demonstrate the use of AllCCS for a number of biological applications and I was impressed with the identification of novel metabolites.

My major comment is that if I have understood AllCCS it is available through a webserver maintained by the group and people need to register for access. While the authors make some data available a lot of what is generated is behind the webserver interface. What are the plans to making this available for the wider community? The database of putative metabolites looks particularly useful. Also, if the tools are made freely available to the community they can be greatly improved in terms of versatility – as an example I think about the developments of xcms by the community. Have the authors considered this? Also, how does the server cope with large datasets as can be generated by ion mobility data?

As a follow up comment while its impressive how well CCS values can be estimated – in many cases its less than 2% - can the authors discuss examples where this is not enough. Some discussion of what can be separated in terms of structural isomers would help. I am thinking of metabolites such as glucose-6-P, glucose-1-P and fructose-6-P which are poorly separated by both chromatography and ion mobility. Some examples of where an annotation can be made but not an identification would be useful.

In addition, I do have some specific comments.

Abstract, line 17. The first sentence should be “the metabolome” rather than just metabolome. I think the paper would benefit from a careful read to fix some of the grammar errors that creep in.

Introduction, line 32. This first sentence is also odd and should be re-written. Similarly for the sentences on lines 54, 58 and 66.

Results. Line 94 – should be laboratories.

Line 98 “As a result, a total of 3,539 unified CCS values were calculated for 2,193 compounds...” I think from what the authors say elsewhere the discrepancy in numbers is down to ionisation mode

rather than different conformers – its probably worth stating this for the reader and the authors are largely annotating one conformer.

Lines 138-141 also need re-writing.

Line 283. Is KEGG the best choice for metabolic reconstructions? Is RECON a better alternative for human metabolism?

A minor point – Student's t-test and student's t-test are written. I think the correct form is with a capital S but needs to be consistent one way or the other depending on whether you believe Student was a genuine author or not!

“All other data supporting the findings of this study are available from the corresponding author on reasonable 602 request.” Should the data be made openly available?

Something is wrong with reference 9.

Reviewer #3 (Remarks to the Author):

The manuscript by Zhu et al. describes a curated atlas of ion mobility collision cross sections (CCS). These collision cross sections are being increasingly used for the purpose of metabolite annotation as a complementary parameter by which unknown molecules, as particularly metabolites in untargeted metabolomics studies, can be better annotated. Although the number of databases with collision cross sections is growing steadily both in number and in size, the dearth of comprehensive CCS databases has prompted many scientists to also look into predicting such CCS values using machine learning approaches. The authors of this manuscript have, in the past, presented several aspects of the work covered in this new manuscript in a number of articles. These include the following: *Anal. Chem.* 2016, 88, 11084–11091 (MetCCS); *Anal. Chem.* 2017, 89, 9559–9566 (LipidCCS); *Bioinformatics*, 33(14), 2017, 2235–2237 (MetCCS) etc. Similar work has also been published by other teams, including: *Anal. Chem.* 2020, 92, 1720–1729; *Anal. Chem.* 2019, 91, 5191–5199; *Anal. Chem.* 2017, 89, 6583–6589; *Chem. Commun.*, 2017, 53, 7624–7627; *Journal of Chromatography A*, 1542 (2018) 82–88; *Analytica Chimica Acta* 924 (2016) 68e76. In that sense, and despite the interesting nature of what is being described in the manuscript, the work is seen as incremental, but not transformative, so publication in *Nature Communications* is not seen as a good option, and a more specialized Journal such as *Analytical Chemistry* or *The Analyst* is recommended.

Response to the reviewers:

The authors would like to thank the reviewers for the helpful comments. We feel these comments have strengthened the manuscript considerably.

Reviewer #1:

Remark to the Author: *“This paper described the creation and validation of an atlas of CCS values. The atlas is created using a combination of experimental CCS (merged from multiple databases) and in-silico predicted CCS. The authors show that they are able to predict CCS better than current systems, and that combining predicted CCS with other data (m/z, predicted MS/MS) improved candidate ranking. The atlas looks to be a resource that has practical utility and the authors do a good job showing that it performs well.”*

Ans: Thanks a lot for the reviewer’s positive comments towards publication.

Comment #1: *“Overall I enjoyed reading this paper. I think it does a good job in setting out the system, evaluating it, and describing the benefits it brings. There are quite a few places in the text though in which more thorough proof-reading is required (too numerous to list here).”*

Ans: Thanks a lot for the reviewer’s comment. We have carefully checked and re-edited the manuscript and supplementary information.

Comment #2: *“The improvement in CCS prediction is good, and the ML methodology looks sound. I would welcome a little more discussion on where the authors think this improvement comes from. Is it simply the larger unified database?”*

Ans: Thank a lot for the reviewer’s comment. We think the improvements of CCS prediction were attributed by three major factors: (1) AllCCS used large and wide-coverage CCS records to train the machine-learning based prediction model, such as numbers of compounds, CCS values, and chemical structural classes; (2) the use of a 5-step standardization strategy and unified CCS values overcome the biases across different instruments and labs and improved the data quality of the training set; (3) the selection of optimized molecular descriptors (MDs) improved the prediction accuracy. As a comparison, MD selection was not performed in MetCCS and DeepCCS. The detailed comparisons were provided in Supplementary Table 7 (see below). The discussion has been added in our revised manuscript.

Supplementary Table 7.**The training set comparison for machine-learning based CCS prediction tools**

Tools	AIICCS	MetCCS	DeepCCS
No. of compounds	1,873	514	1,076
No. of CCS values	2,646	796	1,665
No. of classes	131	52	104
No. of source labs	4	1	4
CCS standardization	Yes	No	No
MD optimization	Yes	No	Not applicable

Note: ISiCLE was excluded from the comparison because it uses a quantum-chemistry based approach for CCS calculation.

Comment #3: “The SVM regression for CCS prediction: the gamma parameters (note: radial basis, not radio basis) seem extraordinarily high (2^8 and 2^{13}). Depending on the way the RBF function is parameterised, this effectively becomes either a linear regression, or something akin to a nearest neighbours scheme. Not necessarily a problem, but something that if I saw it in an analysis, would ring some alarm bells for me. Have the authors looked into this?”

Ans: Thanks a lot for the reviewer’s careful observations and comments. We did not realize this phenomenon during the method development. To evaluate this, we added a comparison between support vector regression (SVR) and multiple linear regression (MLR) using the same molecular descriptors (Supplementary Table 12). The SVR prediction demonstrated better performances than MLR method with validation sets 1 and 2. The results indicated that the optimized parameters in SVR prediction make the model towards a linear regression, but has better performances. The term “radial basis” has been corrected. The results and discussion have been added in the revised manuscript.

Supplementary Table 12.**The comparison between SVR and MLR**

	Validation set 1		Validation set 2	
	SVR	MLR	SVR	MLR
Median relative error (%)	1.67	1.82	2.03	2.94
Mean relative error (%)	2.25	2.60	3.19	3.44
Percentages (error \leq 4%)	85.6	77.5	76.4	66.3

Note: the abbreviations SVR and MLR represent support vector regression and multiple linear regression, respectively.

Comment #4: “When combining with MS/MS, the authors use only in-silico MS/MS on their side. Why is this? in-silico MS/MS prediction is still not great. For the “unknown” experiments, predicted MS/MS is obviously the only route. However, could the authors not have done some experiments with real MS/MS library spectra? Playing devils advocate: perhaps the improvement CCS gives about in-silico MS/MS is because in-silico MS/MS is not reliable. Would we see the same improvements with real MS/MS libraries? There are plenty in the public domain (Massbank, GNPS, HMDB, etc).”

Ans: Thanks a lot for the reviewer’s suggestion. In the revised manuscript, we have added an evaluation to combine the predicted CCS values with experimental MS/MS spectra. The experimental MS/MS spectral library was downloaded from GNPS with a total of 13,499 compounds (<https://gnps.ucsd.edu/ProteoSAFe/libraries.jsp>; accessed on May 23th, 2020). When performing the multi-dimensional match using experimental *m/z*, MS/MS spectra and CCS values towards the predicted CCS values and GNPS MS/MS spectra, the average candidates were significantly reduced from 7.3 to 1.7 (77%) with the addition of CCS match (Supplementary Figure 7). The results demonstrated the CCS match also reduced the candidates and improved the accuracy when combined with the experimental MS/MS library. The results and discussion have been added in the revised manuscript and supplementary information.

Supplementary Figure 7. The candidate reduction of the multi-dimensional match using the predicted CCS values and the experimental MS/MS spectral library. The experimental MS/MS

spectral library was downloaded from GNPS with a total of 13,499 compounds. Other parameters were kept the same as the in-silico MS/MS spectral match in Figure 4.

Comment #5: *“Reproducibility: the authors do a good job of providing the supplementary material to allow another to produce their results. One thing that I cannot find are the in-silico predicted spectra that were used in the experimental section. Recreating these would be challenging (e.g. CFM-ID takes a long time to run) and could prohibit reproduction. I think they should be included as, e.g., one or more .mgf files.”*

Ans: We agree with the reviewer’s comment. All in-silico predicted MS/MS spectra from CFM-ID have been provided in Supplementary Data 5. Please be aware that MetFrag and MS-Finder do not allow to output in-silico MS/MS spectra.

Reviewer #2:

Remark to the Author: *“The manuscript by Zhou and colleagues describes a software tool for the large scale deployment of ion mobility collision cross-section data for metabolite identification of both known and unknown metabolites. While a number of tools have been created previously, including by the senior author, to search databases of known standards and also predict CCS values for metabolites without CCS measures, AllCCS dramatically extends the use of CCS values and ion mobility for metabolite identification and annotation by unifying a number of experimental CCS values to produce consensus values and predicting CCS values for an impressive database of putative metabolites. The authors demonstrate the use of AllCCS for a number of biological applications and I was impressed with the identification of novel metabolites.”*

Ans: Thanks a lot for the reviewer’s positive comments towards publication.

Comment #1: *“My major comment is that if I have understood AllCCS it is available through a webserver maintained by the group and people need to register for access. While the authors make some data available a lot of what is generated is behind the webserver interface. What are the plans to making this available for the wider community? The database of putative metabolites looks particularly useful.”*

Ans: Thanks a lot for the reviewer’s comment. First of all, every user could access AllCCS webserver to view, calculate and download both experimental and predicted CCS values in AllCCS. The prediction algorithm and codes of AllCCS is also available on GitHub.

In order to make AllCCS available for the wider community, a series of collaborations are in progress, including: (1) we are working with the developers from Human Metabolome Database (HMDB) to deploy CCS values into HMDB. In addition, other databases such as DrugBank and FoodDB will also be deployed through the collaboration with the HMDB bioinformatics team;

(2) we are collaborating with Dr. Hiroshi Tsugawa to integrate AllCCS into data processing tool MS-DIAL 4. This allows to accelerate the workflow from raw data processing to compound identification. The related work has been posted on bioRxiv (<https://doi.org/10.1101/2020.02.11.944900>), and accepted by **Nature Biotechnology** (online on June 15, 2020 by the journal).

(3) we are also collaborating with instrument vendors (currently Agilent and Bruker) to deploy AllCCS into vendors’ software. This will allow the easy access to AllCCS for common users.

Finally, as requested by the reviewer, the CCS values for putative metabolites generated from the in-silico reactions have been deposited in AllCCS (named as ExtDB).

The related discussion has been added in the revised manuscript.

Comment #2: *“Also, if the tools are made freely available to the community they can be greatly*

improved in terms of versatility – as an example I think about the developments of xcms by the community. Have the authors considered this?”

Ans: We agree with the reviewer's comment. As we mentioned in the response to Comment #1, we are collaborating and integrating AllCCS into data processing tools such as MS-DIAL 4 and vendors' software tools. This allows to accelerate the workflow from raw data processing to compound identification. For sure, developers could also integrate AllCCS with other data processing software tools such as XCMS and MZmine. The related discussion has been added in the revised manuscript.

Comment #3: *“Also, how does the server cope with large datasets as can be generated by ion mobility data?”*

Ans: Thanks a lot for the reviewer's comment. Currently, AllCCS does not directly process raw IM-MS data files. Instead, we recommend the users processing raw IM-MS data files using MS-DIAL 4, or vendors' software tools (e.g., Agilent Mass Profiler and Waters Progenesis QI). Then, the users could download AllCCS and perform the annotation within these tools. In addition, users can also upload the feature table into the AllCCS server for metabolite annotation. The related discussion has been added in the revised manuscript.

Comment #4: *“As a follow up comment while its impressive how well CCS values can be estimated – in many cases its less than 2% - can the authors discuss examples where this is not enough. Some discussion of what can be separated in terms of structural isomers would help. I am thinking of metabolites such as glucose-6-P, glucose-1-P and fructose-6-P which are poorly separated by both chromatography and ion mobility. Some examples of where an annotation can be made but not an identification would be useful.”*

Ans: We agree with reviewer's comment. Although the CCS prediction has made effective improvements to ~2% prediction errors, the identification of metabolite isomers is still a challenge due to the limit resolution of ion mobility (IM) separation. In the revised manuscript, we discussed the IM separation of 4 metabolite isomers, including glucose-6-phosphate (G6P), glucose-1-phosphate (G1P), fructose-6-phosphate (F6P) and fructose-1-phosphate (F1P) (see Supplementary Figure 18). Four metabolite isomers were analyzed using Agilent DTIM-MS with a IM resolution of 40-60. The results demonstrated that metabolite isomers were poorly separated with CCS differences ranging from 0.6% to 3.4%. Therefore, it is impossible to differentiate the isomer pairs of G6P/G1P and F6P/F1P. In addition, we performed the separation simulations with different IM resolutions. The results demonstrated that the isomer pairs G6P/ G1P and F6P/F1P are partially separated with a IM resolution of 200, and baseline separated with a IM resolution of 500. In summary, although many metabolite isomers are difficult to be distinguished, the annotation accuracy will be continuously improved with the availability of high-resolution IM instruments such as TIMS and cyclic IM. The

results and discussion have been added in the revised manuscript.

Supplementary Figure 18. The IM separation for 4 monosaccharide phosphate isomers. (a) The structures of 4 monosaccharide phosphate isomers: glucose-6-phosphate, G6P; glucose-1-phosphate, G1P; fructose-6-phosphate, F6P; fructose-1-phosphate, F6P; **(b)** the separation of 4 monosaccharide phosphate isomers with Agilent DTIM-MS 6560 with a IM resolution of 40-60; **(c-e)** the simulated IM separations of monosaccharide phosphate isomers using different IM resolving powers (Rp): Rp=60 **(c)**; Rp=200 **(d)**; and Rp=500 Rp **(e)**. Rp was defined as the ion drift time (DT) divided by the peak full width at half-maximum height (FWHM) in DTIM-MS.

Comment #5: “Abstract, line 17. The first sentence should be “the metabolome” rather than just metabolome. I think the paper would benefit from a careful read to fix some of the grammar errors that creep in. “

Ans: This description has been corrected. We have carefully checked and re-edited our manuscript and supplementary information during the revision.

Comment #6: “Introduction, line 32. This first sentence is also odd and should be re-written. Similarly for the sentences on lines 54, 58 and 66.”

Ans: We have rewritten these sentences in the revised manuscript.

Comment #7: “Results. Line 94 – should be laboratories. “

Ans: This has been corrected in the revised manuscript.

Comment #8: *“Line 98 ‘As a result, a total of 3,539 unified CCS values were calculated for 2,193 compounds...’ I think from what the authors say elsewhere the discrepancy in numbers is down to ionisation mode rather than different conformers – its probably worth stating this for the reader and the authors are largely annotating one conformer.”*

Ans: We agree with reviewer's comment. This sentence has been modified as *“As a result, a total of 3,539 unified CCS values with different adduct forms were calculated for 2,193 compounds with definitive confidence levels.”*

Comment #9: *“Lines 138-141 also need re-writing.”*

Ans: These sentences have been modified as *“To the best of our knowledge, AllCCS is the largest and most comprehensive CCS database. All predicted CCS values were specified to confidence level 4, and have been deployed in AllCCS webserver. These records of compounds can be easily retrieved with different identifiers, such as SMILES, InChI, and InChIKey.”*

Comment #10: *“Line 283. Is KEGG the best choice for metabolic reconstructions? Is RECON a better alternative for human metabolism?”*

Ans: Thanks a lot for the reviewer's comment. Both of KEGG and RECON are important and popular databases in biology. RECON curated metabolic reactions and pathways mainly for human species. As a contrast, KEGG is one of the most curated databases covering ~6000 species. More importantly, KEGG covered various compounds, including primary metabolites, secondary metabolites from plant and bacterium, and xenobiotic compounds. In our work, we have to demonstrate the effectiveness of our strategy with different biological species, such as mouse, fruit fly, and human (Figures 4, 5 and 6). It is reasonable to choose KEGG as an example database, and further employ it for metabolic reconstruction. We also agreed that other databases such as RECON are good choices for metabolic reconstruction towards human metabolism study. The related discussion has been added into the revised manuscript.

Comment #11: *“A minor point – Student's t-test and student's t-test are written. I think the correct form is with a capital S but needs to be consistent one way or the other depending on whether you believe Student was a genuine author or not!”*

Ans: This description has been modified as *“Student's t-test”* in the revised manuscript.

Comment #12: *“All other data supporting the findings of this study are available from the corresponding author on reasonable request.’ Should the data be made openly available?”*

Ans: We agree with the reviewer's comment. In the revised manuscript, all raw data files have been

uploaded into MetaboLights (MTBLS1622 and MTBLS1693) for public access. Other related data files were also provided in the Supplementary Data. The source codes of CCS prediction have been uploaded into GitHub (<https://github.com/ZhuMetLab/AIICCS>).

The description has been added in the revised manuscript as: “**Data availability.** *All raw data files can be accessed at MetaboLights (MTBLS1622 and MTBLS1693). The annotation results for all metabolomics datasets were provided in the Supplementary Data 7-8. A reporting summary for this Article is available as a Supplementary Information file.* **Code Availability.** *The source code of AIICCS prediction was provided in Github [<https://github.com/ZhuMetLab/AIICCS>].*”.

Comment #13: “*Something is wrong with reference 9.*”

Ans: The original reference 9 has been removed in the revised manuscript.

Reviewer #3:

Remark to the Author: *“The manuscript by Zhu et al. describes a curated atlas of ion mobility collision cross sections (CCS). These collision cross sections are being increasingly used for the purpose of metabolite annotation as a complementary parameter by which unknown molecules, as particularly metabolites in untargeted metabolomics studies, can be better annotated. Although the number of databases with collision cross sections is growing steadily both in number and in size, the dearth of comprehensive CCS databases has prompted many scientists to also look into predicting such CCS values using machine learning approaches. The authors of this manuscript have, in the past, presented several aspects of the work covered in this new manuscript in a number of articles. These include the following: Anal. Chem. 2016, 88, 11084–11091 (MetCCS); Anal. Chem. 2017, 89, 9559–9566 (LipidCCS); Bioinformatics, 33(14), 2017, 2235–2237 (MetCCS) etc. Similar work has also been published by other teams, including: Anal. Chem. 2020, 92, 1720–1729; Anal. Chem. 2019, 91, 5191–5199; Anal. Chem. 2017, 89, 6583–6589; Chem. Commun., 2017, 53, 7624–7627; Journal of Chromatography A, 1542 (2018) 82–88; Analytica Chimica Acta 924 (2016) 68e76. In that sense, and despite the interesting nature of what is being described in the manuscript, the work is seen as incremental, but not transformative, so publication in Nature Communications is not seen as a good option, and a more specialized Journal such as Analytical Chemistry or The Analyst is recommended.”*

Ans: Thanks a lot for the reviewer’s comment. We agreed with the reviewer that the dearth of comprehensive CCS databases facilitated the development of machine-learning based CCS predictions from our group and others. Compared to other tools, AllCCS has great values and advantages in the following aspects:

(1) AllCCS is the most comprehensive atlas to embrace both experimental and predicted CCS values, and provides the largest platform to store, standardize, and search available CCS resources;

(2) The improved CCS prediction algorithm in AllCCS outperformed other existing tools in terms of the accuracy, coverage and applicability;

(3) The representative structure similarity (RSS) developed in AllCCS could indicate the errors of predicted CCS values, which has not been achieved in other tools;

(4) We systematically demonstrated that AllCCS enabled the multi-dimensional match and substantially improved the accuracy of metabolite annotation;

(5) We demonstrated the use of AllCCS to discover unknown metabolites and reveal the additional chemical and metabolic insights towards biological processes.

Overall, AllCCS has provided a high-quality and unified CCS database for the IM-MS community, and further enabled both known and unknown metabolite annotation in untargeted metabolomics. We believe our work is significant and suitable for the broad interests of scientists in the field.

REVIEWERS' COMMENTS:

Reviewer #1 (Remarks to the Author):

The authors have done a very thorough job of responding to my comments. I particularly welcome the additional results, especially those that show the improved performance with real MS/MS data (which is, I think, a more common use-case).

I would be happy to see the manuscript published now.